# Maize/Soybean Intercropping with Straw Return Increases Crop Yield by Influencing the Biological Characteristics of Soil

**DOI:** 10.3390/microorganisms12061108

**Published:** 2024-05-30

**Authors:** Jingjing Cui, Shuang Li, Bate Baoyin, Yudi Feng, Danyang Guo, Liqiang Zhang, Yan Gu

**Affiliations:** 1College of Agriculture, Jilin Agricultural University, Changchun 131008, China; 2College of Plant Science, Jilin University, Changchun 130012, China

**Keywords:** intercropping, microbial biomass carbon, soil enzyme, soil microbial community, straw returned, yield

## Abstract

With mounting demand for high-quality agricultural products and the relentless exploitation of arable land resources, finding sustainable ways to safely cultivate food crops is becoming ever more important. Here, we investigated the effects of the integrated cropping technique “straw return + intercropping” on the soil aggregates as well as the microbial biomass carbon (MBC) content, enzyme activities and microbial diversity in soils of maize and soybean crops. Our results show that in comparison to straw removal and monoculture, straw return and intercropping increase the rhizosphere’s MBC content (59.10%) of soil, along with urease (47.82%), sucrase (57.14%), catalase (16.14%) and acid phosphatase (40.66%) activities as well as the microbial diversity under maize and soybean. Under the same straw treatment, the yield of maize when intercropped surpassed that when grown in monoculture, with the land equivalent ratio of the intercropping treatment under straw return being highest. Overall, the intercropping of maize and soybean is beneficial for the healthy development of sustainable agriculture in the black soil region of northeast China, especially when combined with straw return to fields.

## 1. Introduction

As a suitable cropping mode for developing sustainable agriculture, intercropping has several key advantages, namely, better crop yield and quality, improved soil nutrient cycling, greater soil fertility, and higher microbial diversity in soil [1,2]. To date, over 100 intercropping combinations have been demonstrated in China alone [3]. Among them, combining Gramineae and Fabaceae (Leguminosae) plants is common with maize/soybean intercropping shown to help in not only improving light energy utilization [4] but also increasing the richness and species number of soil microorganisms while bolstering the available content of soil nutrients and their conversion rates, thereby promoting the aboveground plant growth and development of the target crop [5]. For instance, in their dynamic analysis of an intercropped maize/soybean system, Liu et al. found that intercropped maize had considerably greater α diversity with a higher Shannon’s index at the tassel stage compared to monoculture [6]. Work by Zaeem et al. revealed that intercropping is better at promoting the growth and diversity of active microbial communities in the soil vis à vis monocropping in a maize/soybean intercropping (IMS) experiment conducted in a large field under cold climate conditions [7]. Sucrase is a hydrolytic enzyme that hydrolyzes sucrose to glucose for direct microbial uptake, and it is closely related to soil microbial activities [8]. Relevant studies have shown that sucrase activity in soil was higher than that in monoculture when Brassica napus was intercropped with watermelon, cabbage, and kale, respectively, while the microbial community structure was significantly changed in each intercropping mode [9]. Acid phosphatase converts organic phosphorus into inorganic phosphorus for plant utilization by activating it [10]. It has been shown that intercropping can effectively promote phosphorus activation at low phosphorus levels while reducing phosphatase activity at high phosphorus levels to regulate soil nutrient structure [11]. Previous studies have shown that maize–soybean intercropping significantly increased urease activity in soil layers with different profiles from 0 to 40 cm. The urease activity was significantly correlated with organic carbon and total nitrogen [12]. Catalase, an oxidoreductase widely distributed in soils, mitigates H_2_O_2_-induced oxidative damage to plant roots [13]. As compared to maize monoculture, the maize–soybean intercropping significantly increased soil peroxidase activity [14]. However, most studies on soil enzymatic activities have focused on different soil fertility. The changes of soil enzyme activities and soil microbial diversity by the IMS system under straw return conditions need to be further investigated.

One of China’s main cultivation areas of maize lies is the black soil region in its northeast, whose expanding planted area entails the production of huge amounts of straw, which has led to a grave disposal problem [15]. Widely recognized as a field management practice that can balance economic gain with ecological benefits, straw return can resolve that environmental pollution problem while effectively improving the physicochemical properties and biological characteristics of soil to ultimately increase the overall fertility of soil [16]. Accordingly, how the structural characteristics of soil microbial communities are changed under straw return conditions is drawing much attention from researchers [17]. It is known that straw return can positively affect the soil microbial activity, mycorrhizal growth, fungal mycelia, and microbial secretions [18]. Zhang et al. reported that long-term straw return augmented the diversity of soil bacterial communities and changed the distribution of their dominant taxa [19]. Yang et al. demonstrated that both furrow straw return and rotary tillage straw return could effectively enhance the microbial community structure and fungal diversity of soils [20]. However, at present, there is no clear conclusion at home and abroad as to whether the changes in soil biological properties will increase, decrease, or remain unchanged after the introduction of stover return to the field under the IMS system.

With greater progress in agricultural techniques, an integrated approach that combines different conventional cropping modes with straw return is garnering more attention. The intercropping mode of maize/wheat strips under film cover with straw return can effectively maintain soil moisture and improves a crop’s photosynthetic characteristics, increasing the yield of maize and wheat grains by 13.8–17.1% [21]. The wheat/maize intercropping system under no-till straw cover has a higher soil equivalence ratio, carbon emission equivalence ratio, and water consumption equivalence ratio, which led to the sustainability evaluation index increasing by 13% [22]. Zhou et al. found that straw-covered oilseed rape/Zoysia japonica intercropping could improve the soil’s microenvironment and hydrothermal environment in farmland, enabling a better comprehensive evaluation of its soil nutrients [23]. Although the above studies can well summarize that both the IMS system or straw return have positive effects on soil biological properties, their research conclusions differ, and involved mechanisms need to be further investigated and verified. How much the combination of the two actually increases soil biology and which change in soil biology has a greater impact on yield (corn or soybeans) has not been clearly established nationally or internationally. The effects of straw return on biological characteristics of soil in the IMS system have yet to be thoroughly investigated. Hence, this study has the following aims: (1) to clarify the effects of the straw return + IMS system on the stability of soil aggregates; (2) to investigate the effects of the straw return + IMS system on the soil’s carbon content and enzyme activity; (3) to clarify the main influencing factors and mechanisms by which the straw return + IMS system determines the biological characteristics of rhizosphere soil; (4) to analyze the influence path of the above factors on crop yield. This study provides a timely, empirical reference for the optimization and promotion of maize/soybean intercropping.

## 2. Materials and Methods

### 2.1. Overview of the Experimental Site

This field experiment was carried out from 2020 to 2021 at the Western Experimental Station of Jilin Agricultural University (124°48′ E, 45°08′ N), whose mean monthly rainfall and mean air temperature during the crop-growing season are shown in Figure 1. There is a frost-free period of 135~140 days and a mean annual effective cumulative temperature of 2850 °C. The test soil consisted of black calcium soil. The 0–20 cm depth layer has an organic matter content of 1.40%, with 2.1 g kg^−1^ of total nitrogen and 53.8 mg kg^−1^ of total phosphorus; 75.9 mg kg^−1^ of alkaline nitrogen; 16.3 and 130.2 mg kg^−1^ of fast-acting phosphorus and potassium, respectively; and a pH value of 7.2. All the above measurement methods are described in Deng et al. (2024) [24].

### 2.2. Experimental Design

A two-factor split-zone design was used, with straw return (R) versus straw removal (N) set as the main plot zones. The straw return method applied was deep tillage: After harvesting in the fall of the previous year, the maize straw is cut short (length < 10 cm) with a chopper and evenly dispersed over the field; next, the straw is tilled deeply into the 25–30 cm soil layer with a turning plow and then rototilled and raked with a hydraulic off-set harrow machine or a combined cultivator. This is completed according to the actual situation to attain a ready state for sowing. Within each zone, three cropping mode treatments were set up: maize/soybean intercropping (IMS), maize monoculture (MM), and soybean monoculture (MS), using the ‘Heinong 40’ soybean variety and ‘Hengdan 188’ maize variety (seeds supplied by the Academy of Agricultural Sciences of Jilin Province). Each MM or MS plot was 1170 m^2^, in which 12 rows of maize or soybean were planted, respectively. Each IMS plot was 3510 m^2^, consisting of three strips per plot, where six rows of maize and six rows of soybean were planted. The row spacing was fixed at 65 cm, corresponding to a soybean density of 200,000 plants ha^−1^, and a maize density of 70,000 plants ha^−1^ (Figure 2). 

### 2.3. Sampling

Rhizosphere soil was collected on 29 September 2020 (at maturity) and again on 2 October 2021 (at maturity) for the determination of its microbial carbon content and enzyme activities. Surface soil samples (0–20 cm depth layer, adjacent to the crop on the ridge) were collected when the crop matured in 2021 for the determination of structure of soil aggregates and microbial diversity. When the maize ears were ripe, two rows were harvested from each plot; after threshing all the kernels, the grain weight was measured and converted into yield per hectare based on the harvested area. The moisture content of kernels was measured with a grain moisture meter (PM8188, 50 MHz, Kett, Tokyo, Japan), using three replicates, and the final yield data were expressed as the weight of grains at a 14% moisture content.

### 2.4. Measurements and Methods

#### 2.4.1. Structure of Soil Aggregates

From each cropping mode treatment plot combined with or without straw return, 1.5–2.0 kg of soil was collected. These samples were then air-dried and mixed, and a portion was passed through a set of different-sized sieves having apertures of 2 mm, 1 mm, and 0.25 mm. After each sieving step, the samples were weighed to calculate the percentage of soil trapped, and then a 25 g sample was prepared according to that percentage and analyzed by wet sieve analysis. Soil samples were soaked in a wet sifter with water for 5 min (making sure that water covered the bottom of the sieve) and then shaken for 10 min by a shaker, and the soil from each sieve was collected, dried, and weighed to calculate the percentages [25]. To gauge the stability of soil aggregates, their mean weight diameter (MWD) (Formula (1)) and geometric mean diameter (GMD) (Formula (2)) were used, which were calculated as follows [26]:(1)$$\mathrm{MWD}\;(\mathrm{mm})=\sum_{i=1}^{n}{\bar{X}}_{i}W_{i}$$
(2)$$\mathrm{GMD}\;(\mathrm{mm})=exp\left[\frac{\sum_{i=1}^{n}{W_{i}\ln\bar{X}}_{i}}{\sum_{i=1}^{n}W_{i}}\right]$$
where ${\bar{X}}_{i}$ is the mean diameter of water-stable aggregates of different sizes (mm); $W_{i}$ is the mass percentage of water-stable aggregates differing in size (%).

#### 2.4.2. Contents of Microbial Biomass Carbon (MBC) and Organic Carbon in Soil

The MBC (mg kg^−1^) in soil was measured by chloroform fumigation [27]. Specifically, 20 g of fresh soil samples was fumigated and unfumigated and then extracted with a 0.5 mol L^−1^ K_2_SO_4_ (Sigma-Aldrich, Shenyang, China) solution. The resulting extract was oxidized with concentrated sulfuric acid potassium dichromate, and then MBC (Formula (3)) was determined by titration with ferrous sulfate as follows:MBC = Ec/0.38 (3)
where Ec refers to the difference between the organic carbon content of fumigated and unfumigated soil samples, with 0.38 indicating the proportion of carbon leached from microbes killed by chloroform fumigation.

#### 2.4.3. Soil Enzyme Activities 

A kit was used to extract urease (Macklin, Registry number for the chemical CAS:NONE16585, ELISA kit, Shanghai, China), catalase (Macklin, CAS:NONE16574, ELISA kit, Shanghai, China), sucrase (Macklin, CAS:NONE16598, ELISA kit, Shanghai, China), and acid phosphatase (Macklin, CAS:NONE16587, ELISA kit, Shanghai, China) from each soil sample. Their activity was determined using an enzyme-labeling instrument (Feyond-A300, 0-4 OD, Aosens, Hangzhou, China) at wavelengths of 630, 240, 508, and 600 nm, respectively [28].

#### 2.4.4. Microbial Community in Soil

From each fresh soil sample, 10 g was taken and added to 90 mL of 0.85% sterile saline (NaCl) (Sigma-Aldrich, Shenyang, China) in a triangular flask; then, it was shaken at 250 r/min for 30 min. This was allowed to stand for 10 min, then diluted to 100-fold, and added to 96 wells (150 μL each) of a Biologic ECO microplate. These plates were incubated at 25 °C under constant temperature and protected from light. The duration of incubation was 24, 48, 72, 96, 120, and 144 h; at each time point, the optical density at 590 nm was read by a Biologic Reader analyzer (BH11-01802, Banghong, Deyang, China) to assed the carbon source metabolism diversity of the microbial community [29].

The overall microbial activity was expressed as the average well-color development (AWCD) (Formula (4)). The Shannon–Wiener diversity (H) (Formula (5)), McIntosh evenness (U) (Formula (6)) and Simpson’s dominance (D) (Formula (7)) indexes were used to characterize the metabolic functional diversity of the soil microbial community [30]:(4)AWCD=[(∑C−R)]/n
(5)$$\mathrm{Shannon}\text{–}\mathrm{Wiener}\;\mathrm{diversity}\;\mathrm{index}\;(H):\;H=-\sum P_{i\;}lnP_{i}$$
(6)$$\mathrm{McIntosh}\;\mathrm{evenness}\;(U):\;U=\sqrt{\left(\sum N_{i}^{2}\right)}$$
(7)$$\mathrm{Simpson}’s\;\mathrm{dominance}\;\mathrm{index}\;(D):\;D=1-\sum P_{i}^{2}P_{i}$$
where *C* is the optical density of the medium in each well, *R* is the optical density of each control well, and n is the 31 carbon source types in three replicates on the Biolog-Eco microtiter plate. *Pi* is the ratio of relative absorbance of the *i*-th well to the total relative absorbance of the whole microplate, *Ni* is the value of relative absorbance of the *i*-th well, and S is the number of wells with a color change.

### 2.5. Statistical Analysis

SPSS 22.0 was used for statistical analysis, and ANOVA was used to compare the effects of cropping mode and straw management on biological characteristics of soil and the maize crop yield. After evaluating the significant differences between the sample means via one-way ANOVA, we used Duncan’s test, a post hoc test, to determine which specific group means were critically different from each other. The FDR (False Discovery Rate) for *p*-values was stringently applied to ensure minimal false positives. We used Pearson correlations to evaluate the relationships between soil enzyme activities and MBC. A random forest model was built in R software v4.3.1 (https://www.r-project.org/) (22 March 2020) to clarify the ranked importance of different environmental factors on crop yield.

## 3. Results

### 3.1. Water-Stable Soil Aggregates

Figure 3a,b show the effects of straw returning and intercropping on the percentages relative changes differences of water-stable soil aggregates when cropping with maize and soybean, respectively. Intercropping decreased the amount of water-stable aggregates of ≤0.25 mm and increased those ≥2.0 mm in maize soil under straw return (R) and straw removal (N) treatments. The ≤0.25 mm aggregates decreased by 10.30% and 3.09%, while the ≥2 mm aggregates increased by 24.97% and 12.15% in R-IMS vs. R-MM and N-IMS vs. N-MM, respectively. This indicated a more pronounced effect of straw returning + intercropping on the increase or decrease of the two sizes of aggregates. Nevertheless, the percentage of aggregates in soybean soil differed from that in maize soil. The percentage of <0.25 mm water-stable aggregates was similar across the straw management and cropping practices, ranging from 57.35% to 58.26%. The percentage of 0.25–2.0 mm water-stable aggregates was 6.43–7.05% higher for N than R under the same cropping mode, but the percentage of >2.0 mm water-stable aggregates was 15.72% to 18.79% higher for R than N. Under the same straw treatment, the percentage of >2.0 mm aggregates showed an increase of 2.60–9.17% for IMS over MS. By contrast, for the <0.25 mm and 0.25–2.0 mm aggregates, their percentage respectively decreased by 0.68–1.27% and 1.67–2.25% in IMS compared with MS.

The MWD and GMD of soil aggregates are proportional to soil structure stability. As Figure 3c,d show, with less protective tillage measures, the MWD and GMD of maize and soybean soil decreased. Under the same straw treatment, both MWD and GMD were larger in intercropping than in monoculture. After straw turning, compared with maize monoculture, the MWD and GMD values under maize/soybean intercropping were 14.29% and 21.21% larger, respectively; under the straw removal, the corresponding increases were 18.75% and 24.00% higher. Considering now the contrasting straw treatments with the same cropping mode, straw return had generally significant positive effects on the MWD and GMD of maize/soybean soil: under intercropping, they were 26.32% and 25.81% higher than for straw removal, while under monoculture, they were 31.25% and 32% higher.

### 3.2. Microbial Biomass Carbon (MBC) 

Figure 4a,b show the effects of straw returning and intercropping on MBC contents in maize and soybean soils. For both maize and soybean, the soil MBC content in 2020–2021 was highest in the R-IMS treatment and lowest in the N-MM or N-MS treatment, with the former 60.35% and 57.84% higher, on average, than the latter, respectively (*p* < 0.05). Comparing the same cropping mode revealed that the soil MBC content under straw return was 26.73–39.24% higher than under straw removal. For the same straw treatment, the soil MBC content increased by 15.91–25.82% on average under intercropping vis à vis monoculture. Over time, the soil MBC content rose by 20.71% in 2021 and 12.39% in 2020 under the R-MS treatments only with no significant change in any other treatments.

### 3.3. Soil Enzyme Activities 

#### 3.3.1. Soil Urease Activity (S-UE)

Figure 5a,b show the effects of straw return and intercropping on S-UE activity in maize and soybean rhizosphere soils. With less protective tillage measures, S-UE activity in either crop’s rhizosphere soil decreased in that R-IMS treatment had the highest activity, whereas the N-MM or N-MS treatment had the lowest activity with the former 47.82% greater than the latter on average. Under straw return, intercropping increased the S-UE activity in rhizosphere soil by an average of 7.32% compared with monoculture, while the corresponding increase was almost three times greater under straw removal. This suggested that straw return could bridge the gap between monoculture and intercropping. When using the intercropping mode, S-UE activity under straw return was 21.45% greater than under straw removal, while in the monoculture mode, S-UE activity under straw return was 38.45% higher.

#### 3.3.2. Soil Sucrase (S-SC12) Activity 

Figure 5c,d show the effects of straw returning and intercropping on S-SC12 activity in maize and soybean rhizosphere soil. With less protective tillage measures, the activity of S-SC12 in maize and soybean rhizosphere soils tended to decline, being highest in the R-IMS treatment and lowest in the N-MM or N-MS treatments, with the former 57.14% greater than the latter. The average increase in S-SC12 activity was 19.49% higher for intercropping than monoculture under straw return and 27.36% under straw removal. However, the difference between the R-MS or R-MM versus the N-IMS treatment was the smallest. Using the intercropping mode, S-SC12 activity in rhizosphere soil under straw return was 16.91% greater than under straw removal; however, under monoculture, that S-SC12 activity was 27.36% higher.

#### 3.3.3. Soil Catalase Activity (S-CAT)

Figure 5e,f show the effects of straw return and intercropping on S-CAT in rhizosphere soil. For maize (Figure 5e), there was no significant difference between its intercropping and monoculture on S-CAT activity in rhizosphere soil (*p* > 0.05). Irrespective of cropping mode, the response in S-CAT activity to stray was generally straw return > straw removal, being 133.2% higher in the former than latter, on average. Furthermore, straw return increased S-CAT activity by 21.51% in 2021 compared to 2020, while straw removal did not change it between years. However, for soybean, the pattern in S-CAT activity in its rhizosphere soil (Figure 5f) deviated from that of maize: S-CAT activity in R-IMS treatment surpassed that in other treatments, being 16.14% greater on average overall. In stark contrast, the S-CAT activity was similar between R-MS, N-IMS, and N-MS treatments.

#### 3.3.4. Soil Acid Phosphatase Activity (S-ACP)

Figure 5g,h show the effects of straw return and intercropping on S-ACP activity in rhizosphere soil. Regardless of the cropping mode used, for both maize Figure 5g) and soybean (Figure 5h), the S-ACP activity in their rhizosphere soil was characterized by a pattern of straw return > straw removal, being 71.37% and 33.39% higher in the former than the latter, respectively. Additionally, straw removal led to corresponding crop values of S-ACP activity that were 21.18% and 14.57% lower in 2021 than 2020 while increasing by 6.53% and decreasing by 8.68% in soybean’s soil under the straw removal treatment.

### 3.4. Microbial Community Functional Diversity in Soil

#### 3.4.1. Carbon Source Metabolic Activity of Microbial Community in Rhizosphere Soil

AWCD is an important indicator of the overall metabolic activity of a soil microbial community from the perspective of a single carbon source. Here, the amount of carbon source utilized by microorganisms generally increased with the extension of incubation time. The AWCD of rhizosphere soil fluctuated slowly during 0 to 48 h of incubation and then rapidly rose. As evinced by Figure 6a,b, in the straw return + intercropping treatment, the carbon source utilization capacity of maize or soybean substrate exceeded that of the other treatments. The utilization of carbon source by microorganisms in the rhizosphere soil of the intercropped soybean after straw return was not different between the middle and late stages of intercropping.

#### 3.4.2. The α Diversity of the Microbial Community in Rhizosphere Soil

To investigate the diversity (α diversity) of soil single samples, the Shannon index (Figure 7a), McIntosh index (Figure 7b) and Simpson index (Figure 7c) of the rhizosphere soil microbial community were calculated for each of the maize and soybean treatments. Evidently, among them, the Simpson index value ranged from 0.92 to 1.00, while the Shannon index and McIntosh index had the same trend across treatments: IMS > MS or MM under the same straw treatment. Additionally, for soybean, its Shannon index was on average 11.29% higher than that of maize under the same treatment, being 15.04% higher under straw removal, yet it was only 7.53% higher under straw return. Bacterial community diversity decreased under MS with a significant decrease of 5.00% and 6.59% (*p* < 0.05) at the N3 versus N0 level. The McIntosh and Shannon indexes changed in a similar manner, taking the ranking of soybean > maize; under the same treatment, compared with maize, soybean’s McIntosh index was on average 20.62% greater, being 23.39% higher under straw removal and 17.86% under straw return.

### 3.5. Crop Yield and Economic Benefit

As seen in Table 1, a greater yield of maize was obtained by intercropping than by monoculture under both straw treatments with increases of 18.7% (straw return) and 12.3% (straw removal). Likewise, the yield of soybean intercropped was higher than in monoculture under straw return, but the difference was not significant; in contrast, the soybean yield was about 3.2% higher from the monoculture than intercropping mode when straw was removed. For the same cropping mode, straw return can increase maize and soybean yields, bolstering these by 20.7% and 21.6% (intercropping) and 14.1% and 16.5% (monoculture), respectively, after applying deep tillage to return straw to the field versus its removal. Regarding the compound yield of maize and soybean, when they are intercropped, this was 20.8% higher when straw was returned to the field than without it. The compound economic value of the monoculture maize crop was 7.4% higher under straw return than straw removal, while that of the monoculture soybean crop was 16.5% under straw return than straw leaving. The highest composite economic value was obtained for monoculture maize under straw return: $4545.60 ha^−1^. The land equivalent ratio (LER) has been used elsewhere in agronomic research to determine the relationship between component crops. Here, the LER peaked at 1.10 when using the maize/soybean intercropping (IMS) system coupled with straw return.

### 3.6. Correlation Analysis

#### 3.6.1. Soil Enzyme Activities—MBC

Correlations between the enzyme activity in rhizosphere soils of maize and soybean vis à vis their MBC content were analyzed by linear regression modeling. For the maize or soybean rhizosphere, urease (S-UE) (Figure 8a,e), sucrase (S-SC12) (Figure 8b,f), catalase (S-CAT) (Figure 8c,g) and acid phosphatase (S-ACP) (Figure 8d,h) all had positive relationships with the soil MBC content (*p* < 0.01). Herein, the rank order of correlation strength between rhizosphere soil enzyme activity and soil MBC content was S-ACP > S-UE = S-SC12 > S-CAT for maize, and it was S-CAT > S-UE > S-SC12 > S-ACP for soybean. Overall, the relationships between soil enzyme activities and the MBC content of soil were strongly influenced by crop species.

#### 3.6.2. Ranking the Importance of Environmental Factors on Plant Yield 

The relevance of the environmental factors to the maize (Figure 9a) and soybean (Figure 9b) yield was investigated using a random forest model. This revealed that the strongest impact on maize yield came from the structure of soil aggregates (50.79%). The relative influence of <0.25 mm and >2 mm aggregates was ranked first and third among all environmental factors, with contributions of 25.79% and 16.32%, respectively, while that of the MBC content of rhizosphere soil was 17.37%. Among the soil enzyme activities, acid phosphatase (S-ACP) had the highest relative influence, at 8.95%, while urease (S-UE) and sucrase (S-SC12) had the least influence, both at 3.95%. Among all factors examined, the microbial community in rhizosphere soil had the lowest summation relative influence of 9.21%. 

However, the environmental factors affected soybean yield in a very different way than for maize. We found that soil microbial community diversity (41.18%), followed by soil enzyme activities (38.50%), were the most influential factors shaping the soybean yield. Among all environmental factors, the relative influence of S-ACP in soil was 26.20%, followed by the Shannon index (21.66%) and McIntosh index (14.71%), being lowest for S-UE in soil at 0.80%. The MBC relative influence was 10.43%. Nevertheless, although the greatest influence on maize yield was clearly the structure of soil aggregates, that factor’s relative influence on soybean yield was much lower at only 9.89%.

## 4. Discussion

### 4.1. Effects of Straw Return + Intercropping on the Structure of Soil Aggregates

The high and low contents of water-stable aggregates in soil are closely related to its resistance to impact and erosion [31]. When water-stable aggregates occur at a high content level, the soil is not readily dispersed after being eroded by water, resulting in good soil stability that favors crop growth. It is known that anthropogenic interventions such as the cropping mode can affect the formation of soil aggregates and their stability [32]. For example, Song et al. reported that straw return and intercropping increased the content of water-stabilized aggregates belonging to the 0.5–1.0 mm and 0.25–0.5 mm grain size classes when compared to a single topdressing pattern [33]. This could be attributed to the accumulation of a rich carbon source substantially increasing the content of soil water-stable aggregates after the decomposition of straw [34]. Dou et al. found that a straw return treatment on three soil types—black soil and brown soil—led to a marked increase in their content of 0.25–2.0 mm and 0.25–1.0 mm water-stable aggregates vis à vis control soils lacking straw, with black and brown soils showing greater improvement in organic carbon content and stabilization of aggregates, resulting in higher crop benefits [35]. Meng et al. demonstrated that compared with date palm monoculture, date palm/clover intercropping at a suitable spacing increased the soil’s content of water-stable aggregates >0.25 mm in grain size, and this trend showed a “gradual decrease with soil depth” [36]. 

In our study, there is a greater content of water-stable aggregates of different sizes in the soil after intercropping than after monoculture for both maize and soybean under the same straw treatment, and it is likewise higher with straw return (vs. straw removal) under the same cropping mode. Additionally, among the four treatments, the most significant gain in efficiency was achieved by straw return + intercropping, which could be due to root secretions produced by the root system after intercropping and changes in other microenvironmental factors. Furthermore, straw’s decomposition would release sufficient organic carbon and other nutrients into the soil environment, thus contributing to an improved soil structure.

### 4.2. Effects of Straw Return + Intercropping on the MBC Content of Soil

The soil’s MBC content can effectively decompose its organic matter and promote the cycling and transformation of nutrients as well as help store nutrients in soil [37]. But the MBC content is sensitive to agricultural management practices, which is why it is often used as a key biological indicator for monitoring environmental change [38]. Greater species diversity has beneficial effects on plant–plant interactions, which improves crop productivity as well as increasing belowground C and N stocks, microbial biomass, and crop residues in soil. Returning straw and its decay will release a large amount of nutrients into soil, increasing this pool of resources for microorganisms, which in turn drives shifts in their community dynamics [39]. Previous studies have demonstrated that after straw is returned to a field, the carbon content of its soil microbial biomass increases [40,41,42]. Here, when compared with straw removal, the content of MBC in the rhizosphere soil of maize and soybean also increased after returning straw to the field whether using a monoculture or intercropping mode. The corresponding percentage increase for monoculture exceeded that of intercropping during the jointing to filling period of maize in 2020 and 2021. Under conditions of long-term maize monoculture and continuous cropping, the soil microbial community structure is relatively uniform, but as more nutrients are released into the soil after the straw decays, that original state of dynamic equilibrium is disrupted, and this may greatly promote the growth of certain microbes. However, in the cultivation of corn and soybean intercropping systems, using different crops can have a greater impact on the microbial number, biomass, and population change in rhizosphere soil via interspecific interactions [43]. That view and our findings are consistent with previous work [44], showing that intercropping improves soil health and enhances the metabolic activity of microorganisms.

### 4.3. Effects of Straw Return + Intercropping on Microbial Diversity in Soil

The changes in AWCD reflect shifts in the carbon source utilization capacity and metabolic activity of soil microbes [45]. Here, we also used the Shannon–Wiener diversity (H), Shannon–Wiener evenness (E), and Simpson’s dominance (D) to, respectively, evaluate the richness and evenness of soil microbes and the dominance of some common ones in the communities. Higher AWCD and diversity index values indicate a greater metabolic activity and functional diversity of soil microbial communities [46]. In this study, the treatment consisting of straw return combined with intercropping increased both the AWCD and microbial diversity index (*p* < 0.05). Compared with the positive effects of intercropping or straw return alone, the latter stimulated soil microorganisms more strongly. In particular, in the VT stage of maize, in response to straw return and straw removal, the AWCD of different culture durations under intercropping was on average 8.1% and 31.0% higher than that under monoculture, respectively. For the intercropping and monoculture modes, the AWCD with straw return was 41.5% and 71.1% higher than that without straw, respectively [47]. In general, the soil populations supported by a system that mixes crops are much more diverse than those under monoculture, and it has been pointed out that the community structure is more strongly influenced by the quantity and quality of organic matter than plant diversity. The retention of straw residues from a crop can enhance the soil’s organic matter and improve its structure to increase its overall fertility for farming [48]. However, we find that intercropping affects the microbial metabolic activity and diversity index of soybean’s rhizosphere soil, especially in the middle and late growth stages. In the field area where straw is removed, intercropping is able to increase both the AWCD and diversity index close to levels under monoculture with straw return.

### 4.4. Effects of Straw Return + Intercropping on Enzyme Activities in Soil

Soil enzyme activities are indispensable for organic matter synthesis and degradation in soils and thus serve as an indicator of soil quality, playing critical roles in the decomposition of microorganisms, plants, and animals [49]. In the present research, soil enzymes under maize and soybean crops attained their greatest activity when straw return was combined with intercropping. However, the effects of single intercropping treatment on soil enzyme activity can vary widely [50]. Meta-analysis has shown that intercropping exerts slightly differing effects that depend on the type of enzyme [51], and that impacts tend to be enzyme-specific [52]. In our study, although intercropping does enhance the soil urease, invertase, catalase and acid phosphatase activities under the same straw treatment, this effect (magnitude of increase) is evidently weaker in the rhizosphere of soybean than that of maize. Rudinskiene et al. reported that intercropping increases crop yields and soil nutrient availability, but the soil enzymes’ activity differs little from that of monocultures [53]. According to other research, straw incorporation bolsters the activities of key soil enzymes [54,55].

### 4.5. Effects of Straw Return + Intercropping on Crop Yield

Devising and implementing a reasonable intercropping system is a promising way to effectively use available resources such as light to increase crop yields [56]. There is no doubt that intercropping can effectively increase the yield of certain crop plants [57] and improve the utilization of land resources [58]. We find that the yield of intercropped maize is 18.7% higher than that of monoculture when straw is returned to the field, and it is 12.3% higher in the absence of straw. Although there is research proving that soybean plants and their yield could be adversely affected by the shading caused by intercropping [59], our results nonetheless show that straw return combined with intercropping is capable of increasing the yield of soybean as well, although to a similar degree between intercropping and monoculture. We used six rows maize and six rows soybean that are suitable for mechanized planting and harvesting; a wider distance between the crops’ rows could augment the grain yield of soybean [60]. For either crop in an intercropping system, whether maize or soybean, long-term straw incorporation should lead to better co-efficiency outcomes for increasing crop yield.

## 5. Conclusions

Different planting methods provide differing overall benefits to the target crop. The mean weight diameter (MWD) and geometric mean diameter (GMD) of soil aggregates increased by 14.29% to 21.21%; soil MBC increased by 60.35% and 57.84%, and soil enzyme activities (urease, sucrase, peroxiredoxin reductase, and acid phosphatase) increased by 16.14% to 57.14% under the treatment combination of straw return and intercropping (R-IMS). Soil microbial community metabolic activity and functional diversity increased by 17.86%. Straw return augments the compound yield of intercropped plants, thus increasing the economic and ecological gains.

## Figures and Tables

**Figure 1 microorganisms-12-01108-f001:**
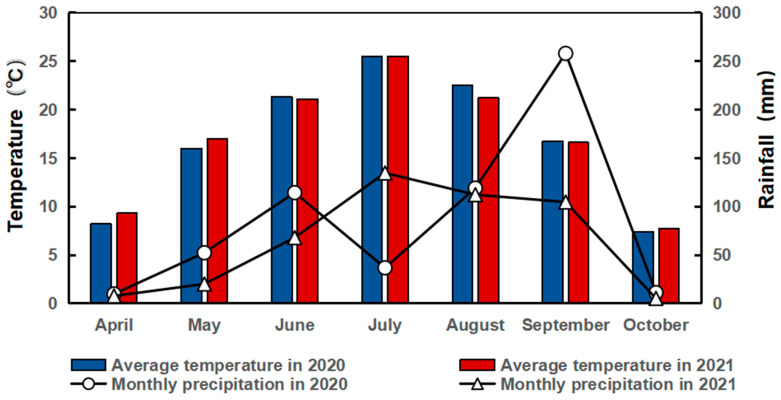
Monthly average rainfall and average temperature during the maize and soybean experiment period (2020–2021).

**Figure 2 microorganisms-12-01108-f002:**
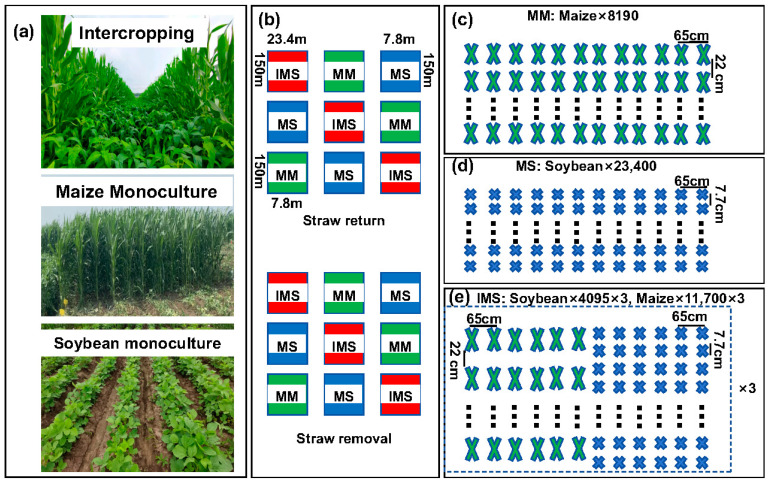
Layout of the experiment’s cropping modes and arrangement of plants. Maize/soybean intercropping (IMS), maize monoculture (MM), and soybean monoculture (MS). In (**a**) is a photograph of the experimental site and (**b**) provides an overview of the experimental design. In (**c**–**e**) is the experimental layout showing the distribution of crops planted in a single treatment, where 8190, 23,400, 4095, and 11,700 are the number of crop plants in the plot; 65 cm refers to the spacing between rows, while 22 cm or 7.7 cm is spacing between plants within a given row.

**Figure 3 microorganisms-12-01108-f003:**
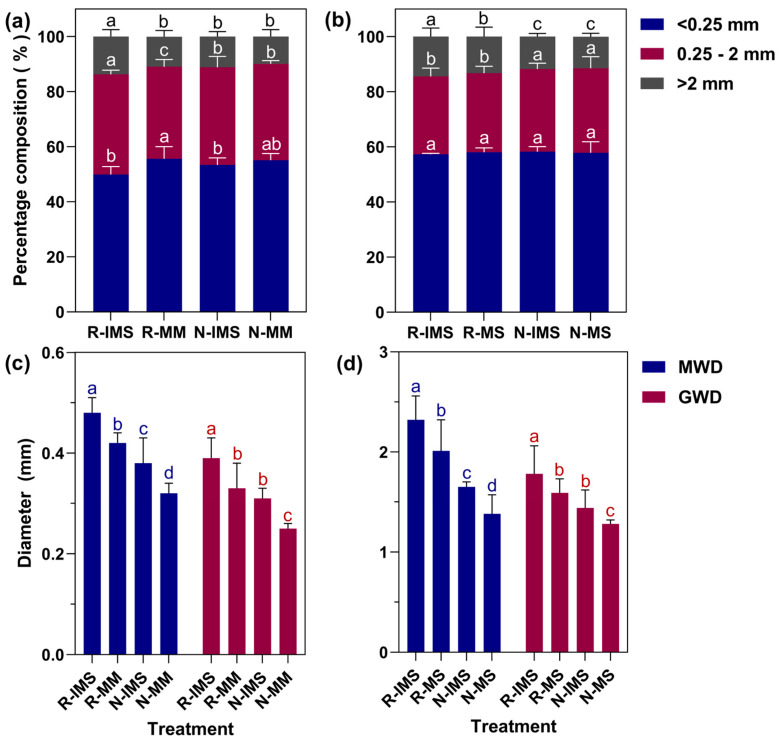
Effects of cropping mode and straw management on the relative distribution and size of maize (**a**,**c**) and soybean (**b**,**d**) soil aggregates. The bars represent the mean ± SE, n = 9 replicates. The a, b, c, and d lettering indicates differences between straw management and cropping practices in the same year and crop (*p* < 0.05). Treatment combinations: straw return (R) versus straw removal (N), maize/soybean intercropping (IMS), maize monoculture (MM), and soybean monoculture (MS), mean weight diameter (MWD) and geometric mean diameter (GMD).

**Figure 4 microorganisms-12-01108-f004:**
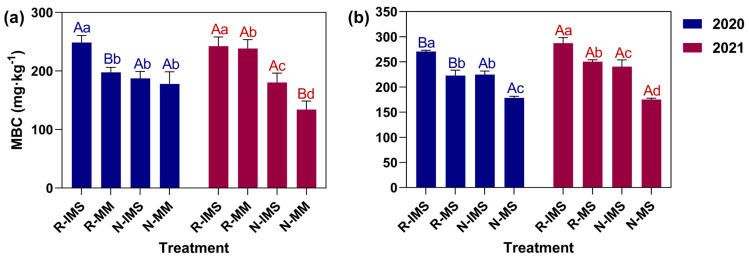
Effects of cropping mode and straw management on the MBC content of maize (**a**) and soybean (**b**) soil. The bars represent the mean ± SE, n = 30 replicates. The A and B lettering indicates differences between years under the same treatment (*p* < 0.05). The a, b, c, and d lettering indicates differences between straw management and cropping practices in the same year and crop (*p* < 0.05). Treatment combinations: straw return (R) versus straw removal (N), maize/soybean intercropping (IMS), maize monoculture (MM), and soybean monoculture (MS).

**Figure 5 microorganisms-12-01108-f005:**
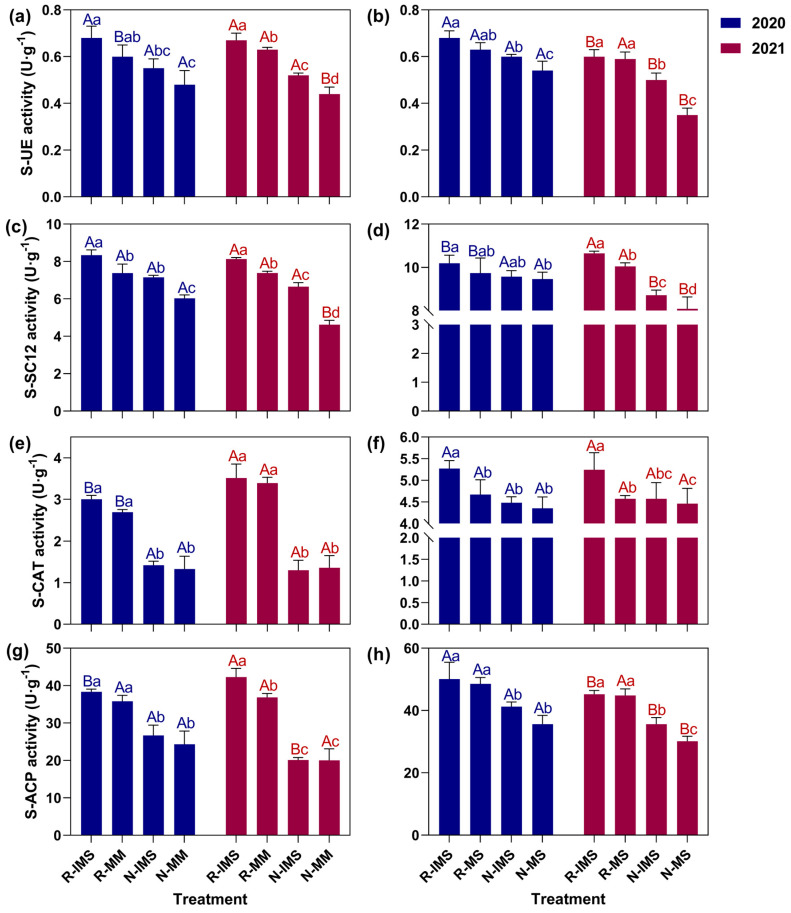
Effects of cropping mode and straw management on urease (S-UE) (**a**,**b**), sucrase (S-SC12) (**c**,**d**), catalase (S-CAT) (**e**,**f**), and acid phosphatase (S-ACP) (**g**,**h**) activity in maize (**a**,**c**,**e**,**g**) and soybean (**b**,**d**,**f**,**h**) soil. The bars represent the mean ± SE, n = 30 replicates. The A and B lettering indicates differences between years under the same treatment (*p* < 0.05). The a, b, c, and d lettering indicates differences between straw management and cropping practices in the same year and crop (*p* < 0.05). Treatment combinations: straw return (R) versus straw removal (N), maize/soybean intercropping (IMS), maize monoculture (MM), and soybean monoculture (MS).

**Figure 6 microorganisms-12-01108-f006:**
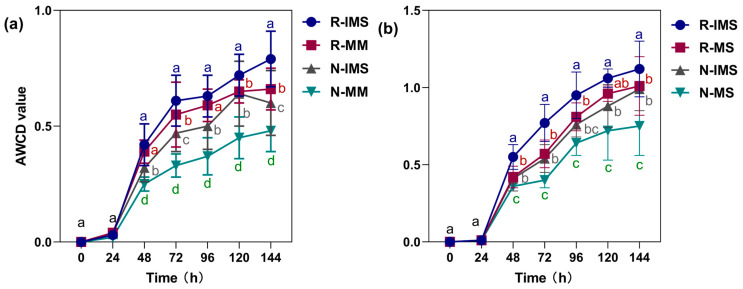
Effects of cropping mode and straw management on maize (**a**) and soybean (**b**) rhizosphere soil microbial community’s carbon source metabolic activity. The a, b, c, and d lettering indicate differences between straw management and cropping practices at the same time point (*p* < 0.05); the symbols indicate the mean ± SE, n = 30 replicates. Treatment combinations: straw return (R) versus straw removal (N), maize/soybean intercropping (IMS), maize monoculture (MM), and soybean monoculture (MS).

**Figure 7 microorganisms-12-01108-f007:**
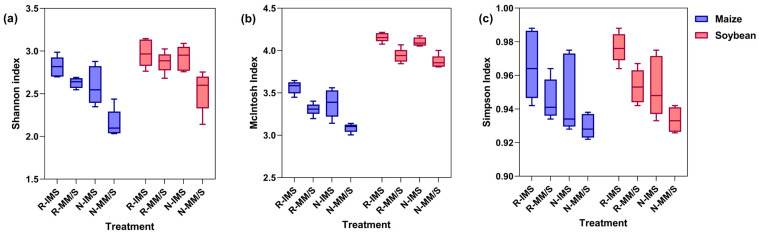
Boxplots showing the effects of cropping mode and straw management on the α diversity of the microbial community in rhizosphere soil. (**a**) is the Shannon index, (**b**) is the McIntosh index, (**c**) is the Simpson index. The boxplots indicate the median and quartiles of the values, n = 30 replicates. Treatment combinations: straw return (R) versus straw removal (N), maize/soybean intercropping (IMS), maize monoculture (MM), and soybean monoculture (MS).

**Figure 8 microorganisms-12-01108-f008:**
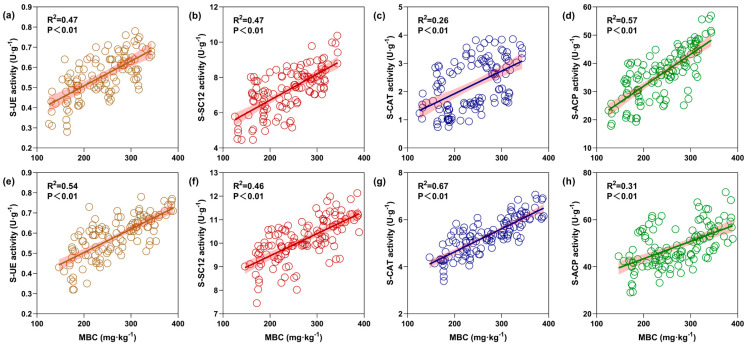
Linear regressions of soil enzyme activities in response to microbial biomass content (MBC). Maize (**a**–**d**) and soybean (**e**–**h**). The shaded part around the fitted regression line indicates the 95% CIs; each circle represent urease (S-UE) (**a**,**e**), sucrase (S-SC12) (**b**,**f**), catalase (S-CAT) (**c**,**g**) and acid phosphatase (S-ACP) (**d**,**h**) in relation to microbial biomass carbon (MBC) content; each panel contains 120 samples.

**Figure 9 microorganisms-12-01108-f009:**
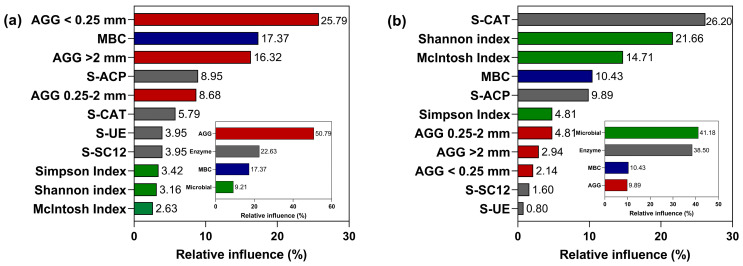
Analyzed importance ranking of environmental factors for their impact on maize (**a**) and soybean (**b**) yield from a random forest model. Soil aggregates (AGG), urease (S-UE), sucrase (S-SC12), catalase (S-CAT) and acid phosphatase (S-ACP) in relation to microbial biomass carbon (MBC).

**Table 1 microorganisms-12-01108-t001:** Effects of straw return on crop yield and LER in the IMS * system.

Treatment		Yield of Maize Strip (kg ha^−1^)	Yield of Soybean Strip (kg ha^−1^)	Composite Yield (kg ha^−1^)			Composite Economic Value (USD ha^−1^)			LER *
				Maize	Soybean	Compound Yield	Maize	Soybean	Compound Value	
Return of straw	Intercropping	14,993.3 a	2594.6 a	7496.6 b	1297.3 b	8793.9 b	2896.63 b	1369.53 b	4266.16 b	1.10 a
	Mono-maize	12,628.8 b	-	12,628.8 a	-	12,628.8 a	4879.66 a	-	4879.66 a	1.00 a
	Mono-soybean	-	2567.8 a	-	2467.8 a	2467.8 c	-	2710.77 a	2710.77 c	1.00 a
No straw	Intercropping	12,425.2 a	2134.1 a	6212.6 b	1067.1 b	7279.7 b	2400.49 b	1766.99 b	4167.48 b	1.05 a
	Mono-maize	11,064.2 b	-	11,764.2 a	-	11,764.2 a	4545.60 a	-	4545.60 a	1.00 a
	Mono-soybean	-	2203.3 a	-	2203.3 a	2203.3 c	-	2325.97 a	2325.97 c	1.00 a

* Land equivalent ratio (LER), maize/soybean intercropping (IMS). The a, b, and c lettering indicates differences between treatments (*p* < 0.05).

## Data Availability

The datasets generated and analyzed during the current study are available from the corresponding author upon reasonable request.

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
