# Peer review of "Maize/Soybean Intercropping with Straw Return Increases Crop Yield by Influencing the Biological Characteristics of Soil"

_microorganisms, 2024, doi:10.3390/microorganisms12061108_

Round 1

Reviewer 1 Report

Comments and Suggestions for Authors

Dear authors,

I suggest that you carefully observe some observations. The study has great potential and addresses a globally relevant topic. However, some points can be improved:

1. In the Introduction section - I suggest the authors improve the bibliographic review in the Introduction, bringing in the most recent studies possible. In the entire article there are only two citations from the year 2023. Therefore, incorporating studies from the years 2023 and 2024 will bring greater impact to this research.

2. In the methodology section - I suggest highlighting in the text the numbering of all formulas/equations present in the article, that is, making the call in the text.

3. In the Discussion section - I suggest that the authors rewrite the sentence from lines 444 to 447.

4. In the Conclusions section, I think it is quite important - The conclusions do not address the quantitative results considered highlights of the article. Therefore, it is important from a comparative point of view, as well as significant correlation results, to present percentage values and highlights in the conclusions.

5. The bibliographic reference of this research has a high impact, however it can be improved with even more recent bibliography.

6. Other simple observations are highlighted in the text, which can be seen in the attached article.

7. Observations regarding the standard standards and simple adjustments: lines 28, 33, 65, 145, 228, 267, 332, 371, 409 and 465 may require corrections and/or adjustments.

Reviewer 2 Report

Comments and Suggestions for Authors

The manuscript shows one-year field experiment with maize and soybean in monoculture and in intercropping as well as with and without straw return to check the influence of such treatments on soil aggregates, soil carbon content, enzyme activity and crop yield. The results are valuable and have practical meaning, however it seems that one season experiment (with only one crop harvest) is a bit short. If you can justify using literature value of such short experiments that will be fine. Moreover, manuscript needs some other improvements. Additional statistical analyses needs to be added. It also seems that not all data which are mentioned/described in the text are presented on figures or in tables. Details are below.   Figure 2 – explain in the caption what is (a), (b)…. and what are the numbers 8190 in the case of maize, 23400 – soybean and 4095, 11700 in IMS. Figure 3 – why in the case of GWD „a”, „b” and „d” letters were used, usually we use consecutive letters, so it should be „a”, „b” and „c” (not „d”) – this is of course not a mistake, but I think it would be better. Lines 182 – 184 „Intercropping decreased the amount of water-stable aggregates of ≤ 0.25 mm and increased those ≥ 2.0 mm in maize soil under straw return (R) and straw removal (N) treatments.”  - In figure 3 there is no statistical analysis of that. Could you perform it for fgure 3a and 3b, as it is in the case of 3c and 3d, as well as for Figure 6 and Table 1. Lines 186 – 187 „This indicated that a more pronounced effect of straw returning + intercropping on the increase or decrease of the two sizes of aggregates” - grammar is here incorrect – verb is missing – this sentence could be „This indicated a more pronounced effect of straw returning + intercropping on the increase or decrease of the two sizes of aggregates”. Line 191 – here you used „significantly” – how do you know that this is significantly? –  did you perform statistical analysis for this? If yes – show the results. Lines 192 – 194 – „Specifically, after straw return via deep tillage, > 2 mm, 0.25–2.0 mm, and < 0.25 mm aggregates were respectively 26.23%, 7.47%, and 6.96% greater in content when soybean was intercropped than when cultivated in a monoculture.” –  where these results are shown? In Figure 3b we have only percentage composition. Line 224 – again how do you know it is no significant – did you perform statistical analysis were year was the factor? If yes show the results of this statistical analysis. Moreover in all figure captions there should be clear explanation to which treatments comparision is performed, for example in figure 4 these letters „a”… show differences only between tratments for specific plant and for specific year? Please make it clear. If you write that something is significantly higher, lower, etc. please suport it with the results of statistical analysis – this remark is for the whole manuscript and for conclusions.   Line 389 – „brown soil” two times – what is the difference between them? – check and correct Comments on the Quality of English Language

english need checking

Reviewer 3 Report

Comments and Suggestions for Authors

The paper deals with examining the combining of intercropping maize and soybean with straw return in terms of enhancing crop yield and soil characteristics. I have no concern about the experimental work completed, nor the quality of the written paper in terms of its structure, methods description and results presentation and discussion. However, I did find many references (new and old) dealing with this topic. Namely, there is a lot of published paper on maize and soybean intercropping, as well as straw return. Hence, my main concern is that I am having difficulty in understanding the novelty, significance of the research. If these two (let's call them) techniques have already proven to have positive effect on the examined parameters separately, according to previous research, then their synergistic effect can only be more positive, as the results of this paper show. Maybe the authors can utilize the introduction section a bit more to emphasize their originality.

Besides this a recommend the following minor corrections for the paper:

1. Line 8: 3442,3 degrees Celsius?

2. Lines 80-83 provide a compositional analysis of the test soil. Tis should be in the results section and discussed compared in the discussion section. In this part the analysis method for the compositional analysis should be described.  If this analysis is a previously published result the it should be referenced.

3. Figure captions of Figures 2, 7 and 9 are missing the desciptions of a), b), c), etc. parts of the figure.

4. Rhizosphere soil samples was at maturity in October, and at what phase were they in September? It is not stated?

5. Line 135: the letter K in kg should not be capital.

6. Line 142: there are two dots at the end of the sentence.

7. Lines 253-255: Contain S-UE, but I think it should be S-SC12?

8. Why is there no data point for 0 h on Figure 6, even though it is mentioned in the text?

Round 2

Reviewer 1 Report

Comments and Suggestions for Authors

Dear authors,

I am satisfied with the main corrections made, which helped to improve the impact of this article.